# Copy number footprints of platinum-based anticancer therapies

**Santiago Gonzalez[1], Nuria Lopez-Bigas[1,2,3]\*, Abel Gonzalez-Perez[1,2]\***

**1** Institute for Research in Biomedicine (IRB Barcelona), The Barcelona Institute of Science and Technology, Barcelona, Spain, **2** Centro de Investigación Biomédica en Red en Cáncer (CIBERONC), Instituto de Salud Carlos III, Madrid, Spain, **3** Institució Catalana de Recerca i Estudis Avançats (ICREA), Barcelona, Spain

\* nuria.lopez@irbbarcelona.org (NL-B); abel.gonzalez@irbbarcelona.org (AG-P)

**Data Availability Statement:** This study uses data previously generated by three main projects: (I) Metastatic tumor cohort data (DR-110_update7) from the Hartwig Medical Foundation (HMF cohort) for academic research upon request (https://www.

## Abstract

Recently, distinct mutational footprints observed in metastatic tumors, secondary malignancies and normal human tissues have been demonstrated to be caused by the exposure to several chemotherapeutic drugs. These characteristic mutations originate from specific lesions caused by these chemicals to the DNA of exposed cells. However, it is unknown whether the exposure to these chemotherapies leads to a specific footprint of larger chromosomal aberrations. Here, we address this question exploiting whole genome sequencing data of metastatic tumors obtained from patients exposed to different chemotherapeutic drugs. As a result, we discovered a specific copy number footprint across tumors from patients previously exposed to platinum-based therapies. This footprint is characterized by a significant increase in the number of chromosomal fragments of copy number 1–4 and size smaller than 10 Mb in exposed tumors with respect to their unexposed counterparts (median 14–387% greater across tumor types). The number of chromosomal fragments characteristic of the platinum-associated CN footprint increases significantly with the activity of the well known platinum-related footprint of single nucleotide variants across exposed tumors.

## Author summary

Chemotherapies, in conjunction with radiotherapy and surgery still constitute the workhorse of the first line of treatment for many tumor types. Some chemotherapies exert their cytotoxic effect through DNA damage, which results, in many instances, in cell death. Particular types of DNA damage associated with certain chemotherapies lead to specific footprints of single nucleotide variants, which have been identified in tumor and healthy cells of exposed people. Here, we demonstrate that platinum-based chemotherapies also leave a type of larger mutational footprints in the genomes of exposed cells. These consist of an increase of chromosomal fragments of copy number between 1 and 4 and length below 10 Mb, possibly resulting from an increased frequency of double strand breaks in the chromosomes of the cells of exposed patients. This increase in structural variants induced by the exposure to platinum-based chemotherapies may have implications for the evolution

hartwigmedicalfoundation.nl/en). (II) Primary tumor cohort data from PCAWG is publicly available upon request (https://dcc.icgc.org/pcawg). (III) Chromosomal segments and clinical data of tumors in the POG507 cohort were obtained from the public domain at https://www.bcgsc.ca/downloads/POG570/. All the code necessary for running the analyses described in the manuscript are available at https://bitbucket.org/bbglab/cn_platinum.

**Funding:** This work was specifically funded by the European Research Council (consolidator grant 682398 to N.L.-B.), the European Union's Horizon 2020 research and innovation program (Marie Skłodowska-Curie grant agreement No. 754510 to S.G.) and the ERDF/Spanish Ministry of Science, Innovation and Universities – Spanish State Research Agency/DamReMap (Project RTI2018-094095-B-I00 to N.L.-B.). The funders had no role in study design, data collection and analysis, decision to publish, or preparation of the manuscript.

**Competing interests:** The authors declare that they have no conflict of interest.

of tumors in treated patients, as well as in the long-term side-effects of chemotherapies they may experience, due to damage accumulated in healthy cells.

## Introduction

Chemotherapies and radiotherapy (in conjunction with surgery) constitute the first line of treatment of many tumor types. In the past half century they have contributed to an important reduction of the mortality related to these malignancies [1]. These treatments are designed to directly damage the DNA, or to interfere with basic processes (such as DNA replication or chromosome separation during mitosis) of cancer cells [2–5]. Exposure to some of them, thus contributes to shaping the genome of cancer cells throughout their evolution. For example, some drugs that directly damage the DNA may, in consequence, leave unique mutational footprints, composed of specific patterns of single base substitutions (SBS) or double base substitutions (DBS) along the genome of exposed tumors [6–11].

The type of DNA damage introduced by some of these anti-cancer therapies may also lead to structural variants in the tumor genome. For example, the exposure to radiation therapy is known to cause the introduction of indels and certain types of translocations [12,13]. However, the footprint of large (starting at the kilobase range) structural variants (SVs) of anti-cancer therapies has not been systematically studied to date. Unlike single base substitutions, hundreds or thousands of which may be contributed to a tumor genome by a mutational process, and the overwhelming majority of which bear no significant consequence to the cell, fewer large SVs occur in a cancer cell. Their lower prevalence is constrained both by their sheer size and by their functional impact and it poses an important hurdle to study the footprint of large SVs that may be associated with anticancer treatments.

Throughout the evolution of tumors, the genome of their cells gain and lose chromosomal fragments of different length. As a consequence, the tumor genome appears like a collage of fragments of the chromosomes of its ancestral normal cell with different copy number and zygosity. One of the most dramatic events affecting the structure of a cell's genetic material is the duplication of its entire genome. This event of whole-genome doubling (WGD) constitutes a hallmark of cancer genome evolution [14–16]. By doubling the size of the genome, the WGD is thought to relax the selective pressure on large SVs, in particular deletions, effectively increasing genome instability and plasticity in the evolution of cancer cells [17–19].

Therefore, we reasoned that metastatic WGD tumors could be exploited to overcome the difficulties outlined above to study the influence of anticancer therapies on the landscape of SVs of tumor cells. We carried out a systematic analysis of 2709 WGD metastatic tumors from patients who, as part of the treatment of their primary malignancies, had been exposed to 85 anticancer therapies, sequenced by the Hartwig Medical Foundation (HMF cohort) [16]. We identified a specific footprint of large loss of heterozygosity (LoH) events associated with the exposure of tumors to platinum-based therapies. A lower prevalence of the same footprints was observed in non-WGD metastatic tumors exposed to the same drugs, prompting us to speculate that WGD may provide tumors with an advantage to withstand their effect.

## Results

### High prevalence of WGD across metastatic tumors

Our starting hypothesis is that WGD tumors exposed to certain chemotherapies accumulate large SVs with specific features. Therefore, we first set out to identify tumors with WGD across

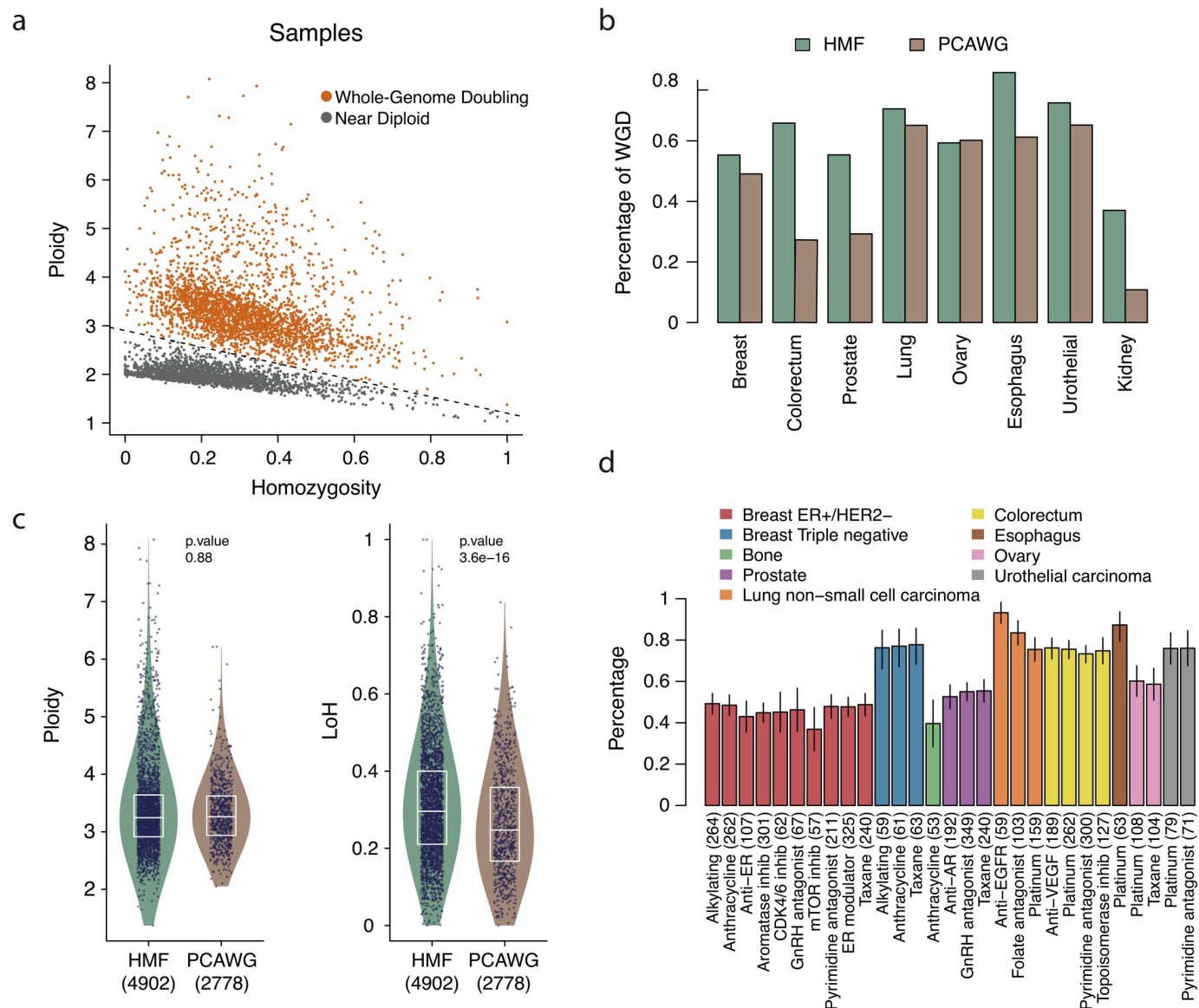

**Fig 1. Features of WGD tumors exposed and unexposed to anticancer therapies.** a) Tumors across one primary (PCAWG) and one metastatic (HMF) cohort may be classified as WGD and non WGD (or nearly diploid) based on their ploidy and level of heterozygosity. b) Different fractions of WGD tumors are observed across different types of cancer in primary and metastatic cohorts. c) Distribution of ploidy (left) and fraction of the genome with LoH (right) of WGD tumors across primary and metastatic cohorts. The boxes inside the violin plots delimit the first, second and third quartiles of the distribution. All tumors in each group are represented as dots. P-values were derived from a two-tailed Wilcoxon-Mann-Whitney test. d) Fraction of WGD metastatic tumors among those exposed to different therapies in the HMF cohort. The number of tumors exposed to each treatment appear in parentheses. The bars represent the mean fraction for tumors in each group (group size below each bar) and the short vertical line, the standard deviation.

two cohorts of cancer samples, one comprising 2778 primary tumors (obtained from the PCAWG consortium; henceforth, PCAWG cohort) [20] and the other 4902 metastatic tumors (obtained from the HMF) [16].

The tumors in these cohorts clearly separate into two clusters when represented in a plane defined by their ploidy and their level of homozygosity (Figs 1A and S1A). These two clusters, which may be separated by a straight line, (as represented in the Figure) group tumors with (orange) or without (gray) WGD. It is known that WGD is more prevalent across metastatic

tumors than their primary counterparts [16]. We corroborated this across different cancer types (Fig 1A; S1 Table).

Moreover, while the interquartile range (that is, between quartiles 1 and 3) of the ploidy of tumors with WGD corresponds to a range between 2.9 and 3.6 (Fig 1A), there are no significant differences between primary and metastatic WGD tumors in the distribution of their ploidy (Fig 1C).

However, as mentioned in the introduction, by relaxing the selective constraints on large SVs, the occurrence of WGD may provide the tumor with an indirect advantage: a more plastic genome. Large chromosomal segments are lost by tumors that experience a WGD, resulting in an average ploidy of 3.2 for both primary and metastatic malignancies (Fig 1A and 1C). However, metastatic tumors bearing WGD exhibit significantly larger fractions of their genome with LoH than their WGD primary counterparts (p-value $3x10^{-16,}$ Figs 1C and S1B). This could be explained as more (or larger) chromosomal segments being lost by WGD metastatic tumors than by WGD primary tumors. We thus hypothesized that the exposure to certain anticancer treatments may leave a footprint of chromosomal fragments with different copy numbers in WGD exposed tumors.

## Profiles of chromosomal fragments associated with anticancer therapies

To test this hypothesis, we leveraged whole-genome sequencing data obtained by the Hartwig Medical Foundation [16] on 1093 WGD metastatic tumors originating from 10 primary sites which had been exposed to 15 families of anticancer therapies (Fig 2A; S2 Table; methods). The most widely used chemotherapy families were platinum-based therapies (498 exposed tumors), pyrimidine antagonists (445 exposed tumors) and taxanes (432 exposed tumors). No apparent differences are observed in the prevalence of WGD across tumors of the same origin exposed to each type of therapy (Fig 1D)

We classified the chromosomal segments in each cancer genome according to their copy number, zygosity and length into 48 groups (Fig 2B) that have been used previously to describe the large-scale organization of tumor genomes with the aim of extracting signatures of copy number variants [21]. Specifically, these features reflect the number of chromosomal segments (ranging in size from less than 1 kilobase to several megabases) with different copy number (CN), between 0 (that is, homozygously deleted) to 9 or more, and their zygosity. For example, a comparison of the two mock examples represented in Fig 2B shows an increase of heterozygous chromosomal segments (across the entire length range) with 3–4 copies in the bottom case, with a complementary decrease of segments with 2 copies and LoH. For simplicity, henceforth we refer to these features as CN categories. If a given pattern of increased and decreased CN categories is significantly associated with the exposure to a treatment family, we can call it the CN footprint of the treatment in the genome, by analogy with previously identified mutational (SBS and DBS) footprints of anticancer treatments [6]. We computed the prevalence of the 48 CN categories across WGD tumors of every cancer type that had been exposed to different treatments. (Tumors of each cancer type were analyzed separately.) Then, we used a Wilcoxon signed-rank test to assess which features significantly differed between tumors (of a given site of origin) exposed and unexposed to a given treatment. We computed one test per CN category, site of origin of the tumor and treatment family, which amounted to 1782 combinations with at least 20 samples in each group (exposed and unexposed). We also computed a fold change between the median prevalence of each category across WGD tumors exposed to the treatments of the family and across those unexposed to them.

We identified 10 CN categories that change significantly (q-value < 0.05) between WGD tumors exposed or unexposed to four types of anticancer therapies (Fig 2C; S3 Table). These

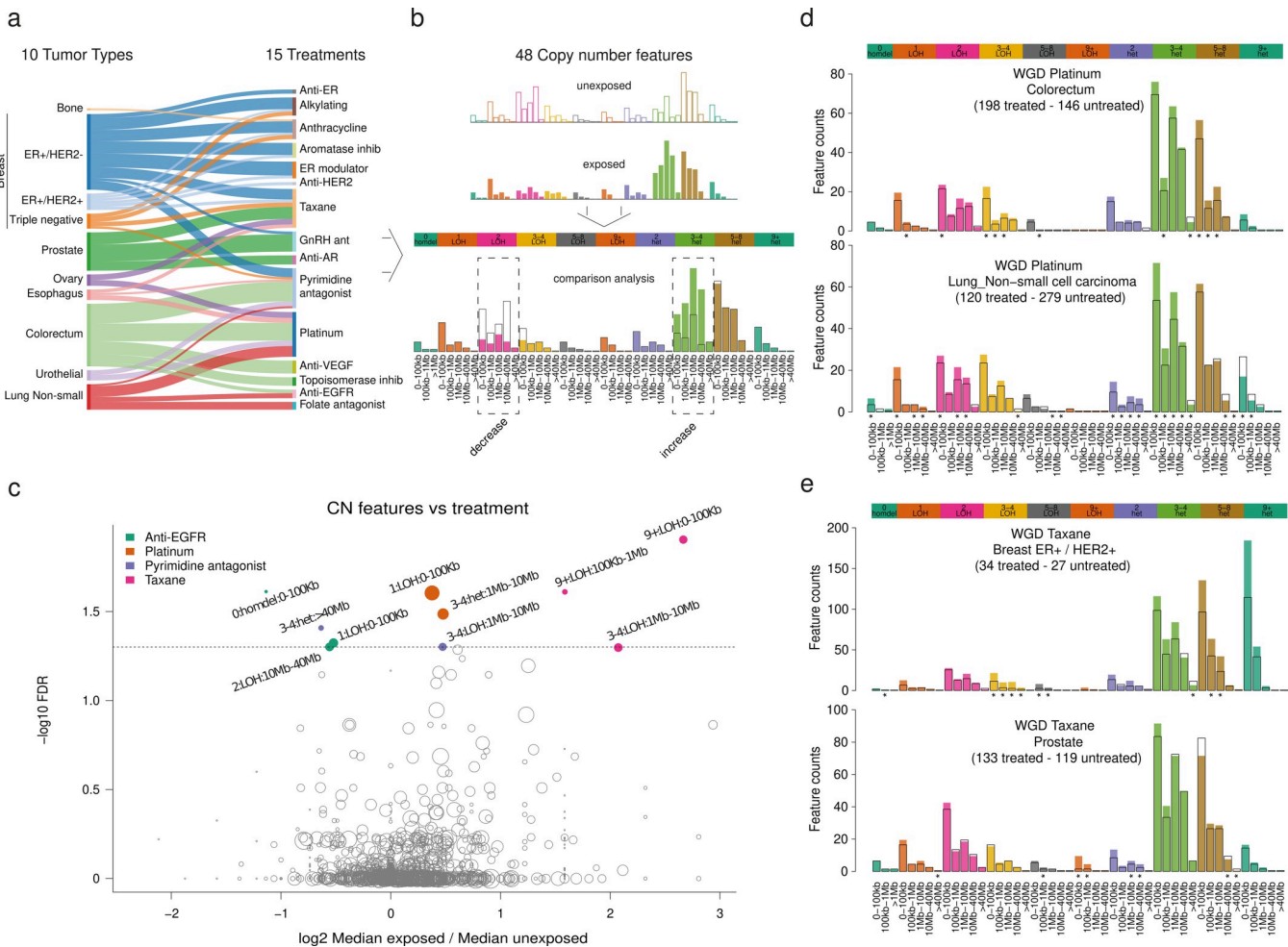

**Fig 2. The profile of chromosomal fragments associated with anticancer therapies.** a) Cancer types (sites of origin) represented in the HMF metastatic cohort and anticancer treatments to which they were exposed. Numbers of tumors of different cancer types in the HMF cohort exposed to anticancer therapies appear also in S2 Table. b) Toy representation of CN categories computed for every tumor and their values for exposed and unexposed tumors (top). Bars represent the count of chromosomal fragments in each CN category. Comparison across CN categories reveals several that appear significantly increased or decreased across exposed tumors with respect to their unexposed counterparts. The CN and ploidy of each category are always shown above the graph, while the range of sizes of chromosomal fragments represented by each bar appear below. For each comparison (one per CN category), a p-value and fold-change can be computed. c) Results (minus logarithm corrected p-value of the Wilcoxon signed-rank test vs fold-change) of 1782 comparisons carried out between CN categories of tumors within a cancer type that were exposed or unexposed to a given anticancer therapy. Significant comparisons (False Discovery Rate corrected p-value below 0.05) d) Results of the comparison of CN categories across WGD colorectal (top) and lung (bottom) tumors exposed or unexposed to platinum-based drugs in the HMF cohort. Significant comparisons (uncorrected p-value below 0.05) are highlighted with asterisks below each graph. e) Results of the comparison of CN categories across WGD breast ER+/HER2+ (top) and prostate (bottom) tumors exposed or unexposed to taxanes in the HMF cohort. Significant comparisons (uncorrected p-value below 0.05) are highlighted with asterisks below each graph.

are platinum-based therapies (across tumors originating in the colorectum or the lung), taxanes (across breast and prostate tumors), anti-EGFR drugs (across lung tumors) and pyrimidine analogs (across colorectal tumors). These associations may be driven by co-variables other than the treatment in question, such as. a second treatment (i.e., two treatments given in combination). Thus, to restrict these results to CN categories change more likely associated to the treatment, we required that: i) the change of CN category continues to be observed when tumors exposed to the second treatment are removed from the comparison; ii) the change of CN cannot be explained by another clinical variable (e.g., sex); and iii) the change is consistently observed across tumor types.

CN categories associated with pyrimidine analogs failed to fulfill the first condition. Specifically, a high fraction of all metastatic colorectal tumors exposed to pyrimidine analogs (N = 194, or 88%) were also exposed to platinum-based therapies. When the comparison is restricted to those exposed only to pyrimidine analogs (N = 26) and those unexposed to pyrimidine analogs or platinum (N = 120), the significance is lost (p-value = 0.23 vs p-value = $2.6\times10^{-4}$ in the comparison for the most significant CN category). Across lung tumors, all 22 WGD samples exposed to pyrimidine analogs are also exposed to platinum-based therapies (N = 120). Overlaps between all pairs of treatments across tumor types are presented in S4 Table. With respect to anti-EGFR therapies across lung tumors, the signal is lost after the overlap with platinum-based drugs (16, 29%) and a gender effect (see Methods) are removed. In detail, the p-value (before correction) of the most significant CN category changes from $6\times10^{-5}$ to 0.034). We thus discarded the associations with anti-EGFR drugs or pyrimidine analogs. In the case of taxanes, CN categories that appear significantly different across exposed or unexposed tumors, were not replicated across tumor types, i.e., different CN categories appear across breast tumors and prostate tumors (Fig 2E). Expanding the analysis to other tumor types treated with taxanes did not yield any common pattern (S2 Fig). It is possible that diverse mechanisms underlie these observed differences across tumor types. It is also plausible that more numerous cohorts of exposed tumors are required to understand whether the observed differences are indeed robust. We also decided to discard the case of taxanes for further analysis within this work.

Unlike the previous cases, the associations between CN categories and the exposure to platinum-based drugs fulfill the three conditions set to consider them significant. Specifically, 11 CN categories–representing diploid LoH fragments below 100 Kb, heterozygous fragments smaller than 100 Kb and larger than 10 Mb, heterozygous fragments with copy number 3–4 smaller than 40 Mb, heterozygous fragments with copy number 5–8 larger than 10 Mb and heterozygous fragments with higher copy number smaller than 1 Mb–appeared significantly increased across 98 lung tumors solely exposed to platinum-based drugs (uncorrected p-value; S4 Table). Moreover, 65 WGD ovarian tumors, which were exposed only to carboplatin (S4 Table) exhibit differences (with respect to their unexposed counterparts) in the same CN categories, although these do not reach significance (S6 Fig; S2 Table).

In summary, CN categories significantly associated with the exposure to platinum-based therapies are detected even in the absence of another treatment, and we are able to observe them across cancer types (see below and S6 Fig). In the next section, we describe and discuss in detail this CN footprint of platinum-based therapies.

## The CN footprint of platinum-based therapies

To study the CN footprint of platinum-based therapies we first analyzed all CN categories with a p-value < 0.05 (before FDR correction) across colorectal and lung tumors (Fig 2D). These CN categories comprise fragments shorter than 40 Mbases across the spectrum of copy number, but mostly heterozygous fragments with CN between 3 and 8 and LoH fragments with CN 3–4 that appear significantly increased across exposed metastatic WGD tumors. Conversely, heterozygous chromosomal fragments longer than 40 Mbases with 3–4 copies appear significantly decreased across exposed tumors originating in either organ (in lung tumors, also those with 5–8 copies).

To better capture this CN footprint of platinum-based therapies, we summed the number of fragments in each CN category in platinum-exposed and unexposed WGD tumors across cancer types, and computed the CN category-wise difference. The CN footprint of platinum-based therapies thus appears as an increase in the number of heterozygous and LoH

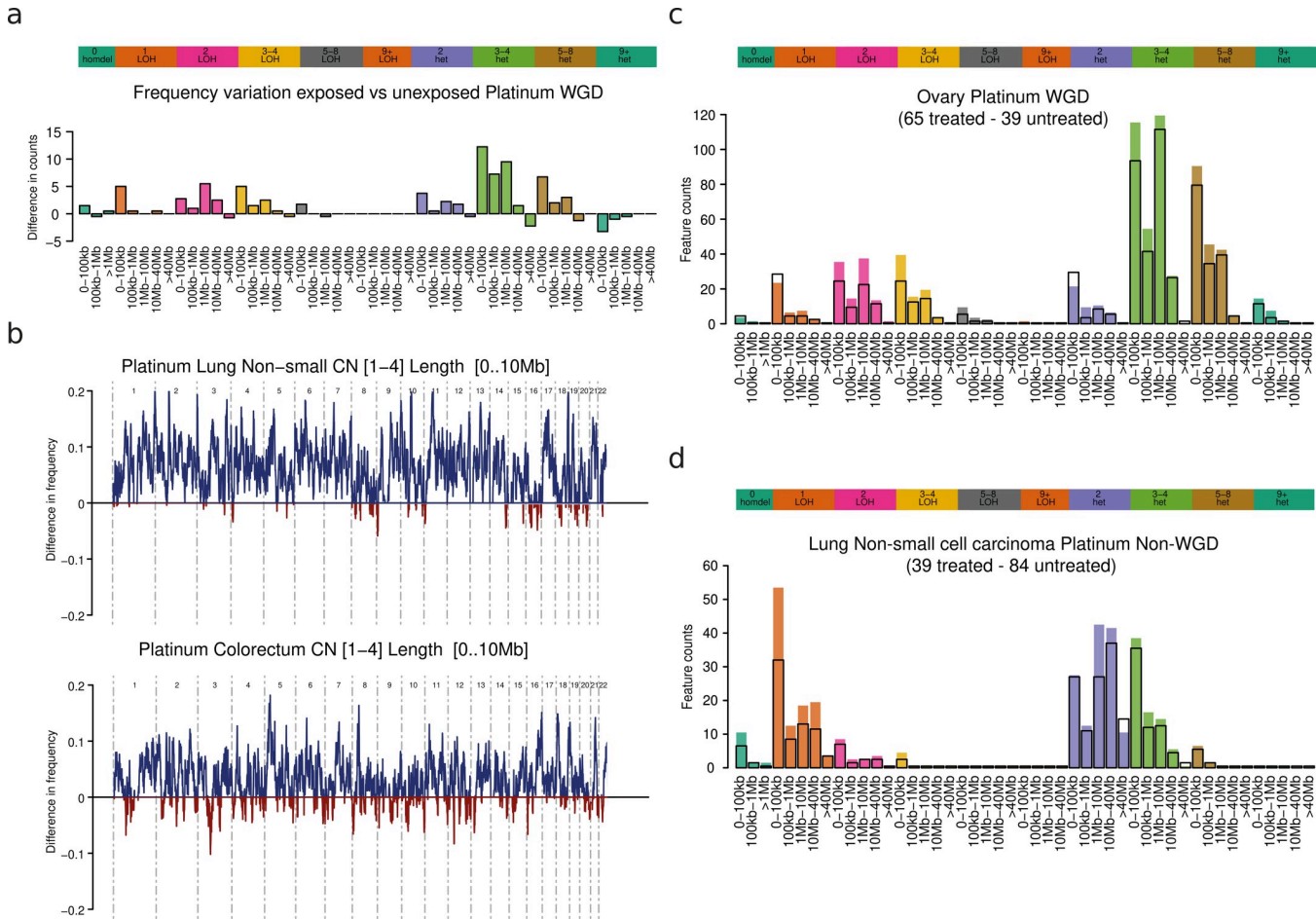

**Fig 3. CN footprint of platinum-based therapies.** a) Representation of the CN footprint of platinum based therapies. The bars represent the average (across WGD lung and colorectal tumors) number of chromosomal fragments in each CN category. b) Frequency of occurrence of chromosomal fragments with copy number 1–4 and length smaller than between 10 Kb and 10 Mb across WGD lung (left) and colorectal (right) tumors in the HMF cohort of tumors exposed (blue, positive) or unexposed (red, negative) to platinum-based drugs. c) Results of the comparison of CN categories across WGD ovarian tumors exposed or unexposed to platinum-based drugs in the HMF cohort. d) Results of the comparison of CN categories across non-WGD lung tumors exposed or unexposed to platinum-based drugs in the HMF cohort.

chromosomal fragments of copy number 1–4 and size smaller than 10 Mb across platinum-exposed WGD tumors (Fig 3A). This footprint is better captured by this array of CN features than by CN signatures identified de novo using a state-of-the-art extraction method or by previously extracted CN signatures across primary tumors (S3A–S3C Fig). This is probably due to intrinsic hurdles in defining signatures of CN, in contrast with commonly defined SBS (or DBS) signatures [21].

Platinum-based drugs (cisplatin, carboplatin and oxaliplatin) are bifunctional agents that create adducts in the DNA by establishing covalent links with one or two closeby bases. Adducts connecting two bases may be intra or interstrand [22]. Although the latter account for less than 15% of all adducts, they may impose a physical obstacle to the separation of the two strands that is required for DNA replication and transcription [23,24]. While some of these interstrand bridges may be repaired by a combination of translesion synthesis and nucleotide excision repair, others result in double strand breaks created by the Fanconi Anemia (FANC) complex [24]. The resulting double strand breaks may then be solved via the homologous recombination (HR), the non-homologous end-joining (NHEJ) repair or other DNA

repair pathways [23–25]. The activity of both, HR and NHEJ in the repair of double strand breaks can result in the deletion of chromosomal fragments of varying sizes. In particular, the use of non-sister chromatids as template for HR repair may lead to loss of heterozygosity [23–26]. Moreover, other mechanisms of repair of double strand breaks which are more error-prone, such as single strand annealing or alternative end joining may also result in the introduction of short insertions and deletions around microhomology elements [24].

Since intrastrand links generated by platinum-based drugs are expected to distribute randomly along the genome of WGD tumors during exposure, we expected that chromosomal fragments resulting from their repair would also distribute evenly across the genome. To test this hypothesis, we computed the difference in the number of chromosomal fragments of copy number 1–4 and size smaller than 10Mb between WGD tumors exposed and unexposed to platinum-based drugs along each chromosome. As anticipated, we observed that chromosomal fragments smaller than 10 Mb (representative of the platinum CN signature) are evenly distributed along the genomes of WGD colorectal and lung tumors (Figs 3B; S4A and S4B). Had this increase in the number of platinum-related chromosomal fragments across exposed tumors been constrained to one or few genomic regions, it would point to positive selection of one or more resistance-associated genes. A recent systematic analysis of the HMF cohort revealed that only mutations in TP53 across stomach adenocarcinomas appear as a potential bona fide driver event associated with the exposure to platinum [27].

Next, we asked if an imbalance between insertions and deletions occurred during the error-prone repair of these interstrand links. To answer this question, we looked at the overall ploidy of platinum exposed and unexposed WGD tumors, and observed no significant differences between these two groups of cases, with the notable exception of lung tumors (S5 Fig). This supports the notion that across colorectal, esophageal, urothelial and ovarian tumors no large imbalances between insertions and deletions occur in the process of repair of platinum inter-strand links. While purifying selection on likely damaging chromosomal deletions could in principle limit their observed number in diploid genomes, this effect is likely removed from this analysis through the study of WGD tumors. Further analyses with larger cohorts are required to clarify this point, which appears especially to understand whether a significant imbalance in favor of deleted chromosomal fragments does occur across platinum-exposed lung tumors.

We then hypothesized that the same CN footprint will be apparent across WGD tumors from other primary sites, even if they don't reach significance due to fewer numbers of exposed samples. Indeed, the same trends of increase of LoH and heterozygous chromosomal fragments smaller than 10 Mbase observed across WGD tumors of ovarian (Fig 3C), esophageal or urothelial (S6 Fig) origin.

While the occurrence of the WGD results in a decrease of the selective pressure on chromosome fragmentation induced by platinum-based drugs–via repair of interstrand bridges–, we expect that such fragmentation also occurs across tumors without WGD. We thus focused on non-WGD tumors exposed to platinum-based therapies. As with WGD tumors, a trend towards the increase of chromosomal fragments below 10 Mb is observed across non-WGD metastatic lung tumors (Fig 3D). Interestingly, this CN footprint is shifted towards smaller CN values (i.e., showing the highest increase for LoH fragments with CN 1; (Fig 3D). Less starting DNA in non-WGD tumors, and the stronger purifying selection resulting from the diploid rather than tetraploid genome are likely the reasons behind this shift towards fragments of smaller CN. However, the overall higher rate of chromosomal fragments across platinum-exposed tumors appears significant across both WGD and non-WGD tumors (Fig 4A). Specifically, the median number of chromosomal fragments of size smaller than 10 Mb across

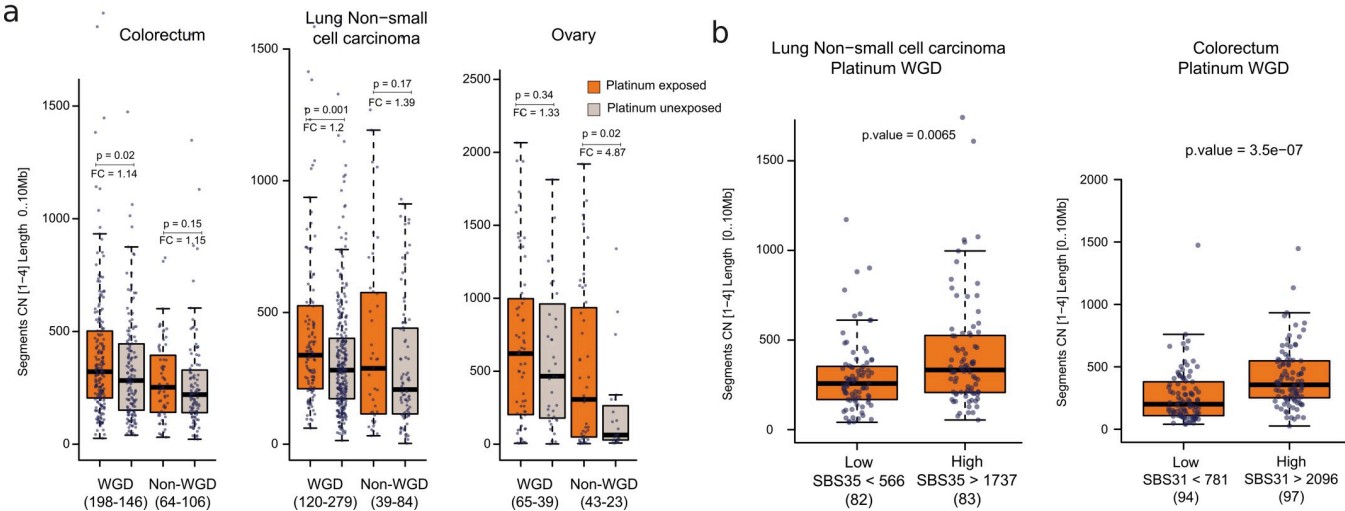

**Fig 4. Relationship between the intensity of the treatment and CN footprint.** a) Distribution of the number of chromosomal fragments with copy number 1–4 and length smaller than 10 Mb across WGD and non-WGD colorectal (left), lung (center) or ovarian (right) metastatic tumors exposed or unexposed to platinum-based drugs. The p-values were derived from a two-tailed Wilcoxon-Mann-Whitney test. Fold-change values (median number of CN fragments of size smaller than 10 Mb across exposed tumors vs its median across unexposed tumors) are shown. Boxes in the boxplot represent the limits of the first and third quartile of the distribution and the whiskers extend between its minimum and maximum, excluding the outliers. Tumors in each group are represented as dots. b) Distribution of the number of chromosomal fragments with copy number 1–4 and length smaller than 10 Mb across WGD lung (left) and colorectal (right) metastatic tumors in the lower or upper tertile of activity of the platinum SBS mutational signature. Boxes in the boxplot represent the limits of the first and third quartile of the distribution and the whiskers extend between its minimum and maximum, excluding the outliers. Tumors in each group are represented as dots. The p-values were derived from a two-tailed Wilcoxon-Mann-Whitney test.

exposed tumors is between 14% (colorectum WGD tumors) and 387% (ovarian non-WGD tumors) greater than the median of their counterpart unexposed samples.

To validate the platinum CN footprint we tapped onto an independent smaller cohort of advanced tumors (POG570) [10]. The POG570 cohort has important differences with respect to the HMF cohort, the most important of which are the smaller sample size and the shorter time elapsed between the end of the treatment and the biopsy of the metastasis (S7A Fig), which is known to decrease the detection of the platinum related mutations through bulk sequencing [11]. Nevertheless, a significant increase in the number of chromosomal fragments below 10 Mb was indeed apparent across 144 (31 platinum exposed) breast and 42 (30 exposed) lung tumors (S7B Fig). The increase of these short chromosomal fragments, however, was not significant across 65 colorectal tumors (40 exposed).

We also reasoned that the more intense the exposure to platinum-based therapies, the greater the observed CN footprint will be. We previously demonstrated that higher activity of the platinum-related mutational (SBS and DBS) footprint results from longer times of exposure to these drugs [6]. We thus reasoned that the activity of this mutational footprint can be used as a proxy of the intensity of the exposure to platinum-based therapies. We found that the rate of chromosomal fragments smaller than 10Mb increases significantly with the increase of the activity of the platinum mutational footprint across metastatic colorectal and lung tumors (Fig 4B).

In summary, we postulate that a platinum CN footprint characterized by increased number of short (below tens of megabases) LoH and heterozygous chromosomal fragments may result from the increase in the number of double strand breaks stemming from platinum-induced interstrand bridges. While it is possible that the presence of this CN footprint increases the aggressiveness of platinum-exposed tumors, the mortality curves are not significantly distinct, S8 Fig). Therefore, further analyses with larger cohorts will be needed to clarify this.

## Discussion

Identifying and measuring the genetic footprints of chemotherapies in exposed cells makes an important contribution to the study of their effects on both tumor and normal cells. We recently tapped on our ability to detect the mutational footprint of several chemotherapies to estimate their mutational toxicity on the basis of regularly used treatment schemes [6,11]. We also studied this toxicity in vivo in cells that were normal at the time of exposure to the chemotherapies. Specifically, we demonstrated that in the emergence of tumors that are secondary to the exposure to chemotherapy this exposure precedes the start of the clonal expansion that gives rise to the secondary malignancy [11]. These studies were made possible by our previous identification of the mutational footprints of these anticancer treatments.

Here, we identify the CN footprint of platinum-based drugs in metastatic tumors, as a result of the exposure to these drugs. The CN profile of platinum-based drugs appears as an increase in the fragmentation of the tumor genome, likely due to the gain of double strand breaks resulting from interstrand links. A clear connection between the ability to repair double strand breaks via homologous recombination repair and the effectiveness of platinum-based therapies has been shown by recent studies that segment patients on the basis of the observed activity of inferred chromosomal instability signatures [28,29]. Furthermore, we found that there is a direct relationship between the intensity of the platinum-based treatment and the degree of the footprint signal.

Though these footprints are apparent across tumors originating in different organs, and across both WGD and non-WGD tumors, their signal is still weaker than that of SBS mutational footprints [6,7,30,31]. The smaller number of CN fragments–orders of magnitude below that of SBS–available to detect the signal is probably the main reason behind this. It is interesting to point out that platinum-based and taxane therapies (which come out significant from the CN categories analysis) are two of the most abundant across the cohort of patients studied. Larger cohorts will thus be needed to corroborate these CN footprints and to potentially detect others associated to different anticancer treatments which may still be below the statistical power afforded by the HMF cohort. The relatively small number of tumors of different origins exposed to platinum-based drugs has also prevented us from exploring in more depth the influence of these treatment families on the aggressiveness of the resulting metastatic tumors, although at least platinum-based drugs appear to increase them. If this trend is confirmed in larger cohorts, this could be important to decide further rounds of treatment on relapse patients.

Since chemotherapies are systemic treatments, their effects–not only the immediate ones, but also those occurring at longer terms–may also affect normal cells of the exposed person. We recently showed that the platinum-related mutational signature is detectable in hematopoietic cells that were normal at the time of exposure to the treatment, and which subsequently suffered a full clonal expansion to produce a full blown acute myeloid leukemia [11]. It seems reasonable to assume that as a result of the increased number of double strand breaks resulting from the exposure to these drugs, normal cells may also suffer an increased number of SVs. Moreover, these SVs–as described by the platinum CN footprint–are bound to affect much larger genomic portions than platinum-contributed point mutations [6,11]. Therefore, their impact on exposed healthy cells and on the development of late effects of the chemotherapy could potentially be greater than those caused by previously recognized platinum-related SBS footprints. Identifying these SVs–a task that may be supported by the CN footprint described here–and understanding if they have a role in the emergence of these secondary malignancies is thus an outstanding goal.

## Methods

### Collection of genomic and clinical data

Somatic copy number (CN) per chromosomal fragment (CN fragments) calculated from whole genome sequencing (and the clinical data for each patient) were obtained from 2 different studies. We obtained primary tumors from the Pancancer Analysis of Whole-Genomes (PCAWG [20]) upon request as explained in https://docs.icgc.org/pcawg/data/. A cohort of metastatic tumors was obtained upon request from the Harwtig Medical Foundation (HMF [16]); https://www.hartwigmedicalfoundation.nl. The PCAWG cohort comprises 2778 whole-genome samples from 47 different primary tumor types. Similarly, HMF comprises 4902 whole-genomes of metastatic tumors across 40 primary anatomical sites. We also obtained single base substitutions (SBS) across HMF tumors.

For each of the samples, genome ploidy represents the mean of the total ploidy of each of the called CN fragments weighted by their genomic length. The fraction of the genome under a loss of heterozygosity (LoH) in PCAWG is computed as the sum of the length of all CN fragments with a minor copy number allele equal to zero divided by the fraction of the genome with discernible CN (i.e., with a value provided by the calling algorithm). In the case of HMF, the LoH estimation is equivalent, but since the CN is not provided as an absolute number, but rather as a continuous value, LoH segments are those with a minor CN smaller than 0.5.

The clinical data obtained for tumors in the HMF cohort included the primary cancer site and the information regarding the pre-treatment (treatment in this manuscript) received by each patient as part of the management of their primary malignancy. This treatment data (see S2 Table) was used as the basis for the identification of CN categories (see below) associated with the exposure to anticancer therapies.

### Whole-genome doubling status and treatments comparison

The whole-genome doubling status (WGD) of each sample is determined based on the average ploidy of the genome and its fraction of LoH [15]. In both cohorts, primary and metastatic, the samples cluster into 2 distinct groups, which allows a clear differentiation between mostly diploid samples and those carrying a WGD (Figs 1A and S1). To perform the classification we have employed the rule:

$$WGD : \quad 2.9 - 1.7 * LoH \quad \quad <= \quad Ploidy.$$

As a representation of a variety of tumors and treatments, we selected all the combinations of tumor types and treatment comprising at least 50 samples (S2 Table) in order to investigate the impact of the treatment on the fraction of WGD tumors in a cancer type or exposed to a given treatment. For each pair tumor-treatment, the percentage of samples carrying a WGD was calculated and the confidence intervals were estimated using the 5 and 95 percentiles of the binomial distribution estimated based on their prevalence and cohort size.

### Selection and evaluation of CN features

This study uses a total of 48 CN categories that classify each chromosomal CN segment according to its absolute copy number (0, 1, 2, 3–4, 5–8, or 9+), zygosity (homologous deletions, heterozygous or loss of heterozygosity), and length (0-100Kb, 100Kb-1Mb, 1Mb-10Mb, 10-Mb-40Mb, or 40Mb+) following the classification implemented by Christopher D. Steele and colleagues [21] (Fig 2B). The CN categories present no overlap, and all chromosomal segments in a genome are classified into one of them.

We filtered out samples with no treatment record and those tumor type-treatment combinations with less than 20 WGD cases. As a result, we obtained a total of 15 treatments and 10 tumor types based on their primary location. The analysis results in 39 combinations of tumor type-treatment to interrogate the impact of anticancer therapies on the selected CN category. For each CN category across each tumor type-treatment group, an independent Wilcoxon's signed-rank test was used to estimate a p-value of the difference of the prevalence of the CN category between exposed and unexposed WGD tumors (S3 Table).

In order to avoid the overlap present across many tumor-treatment pairs, only one treatment is selected for the multi-test correction from those with an overlap of more than 90% of the samples in either treated or untreated groups. All obtained values were corrected by the effect of the multiple tests using the Benjamini & Hochberg false discovery rate.

We performed a manual curation of significant results to determine the appropriate treatment for those considered as overlapping (see previous paragraph). For colorectal cancer, pyrimidine analogs and platinum cohorts present an overlap of 88 and 97% of cases respectively, and show significant associations between some CN categories and the treatments (S4 Table). We selected platinum as the main contributor due to the consistency in treatment footprint across other tumor types, including some with less or none exposure to pyrimidine analogs (Figs 2, 3 and S5). Similarly, anti-EGFR drugs lose their significance once the overall with platinum and a sex bias effect are accounted for.

## Copy number signature extraction

With the number of the 48 CN categories in each tumor (i.e., their CN feature profile), we first generated a matrix of counts using the SigProfilerMatrixGenerator version 1.1. Then, CN signatures were extracted using the sigProfilerExtractor version 1.1 (Islam et al., 2020) and the default set of parameters. The solution recommended by the tool based on signature stability and the average error of reconstruction of the CN profiles comprised 9 CN signatures. To determine if any of these CN signatures was associated with the exposure to any treatment, we carried out a Wilcoxon signed-rank test of the activity of each signature (also provided by the sigProfilerExtractor as part of the CN signatures extraction de novo) across exposed and unexposed WGD tumors, for each tumor type separately (S3A Fig).

Then, we analyzed (part of the sigProfiler package) the equivalence between these 9 CN signatures and the CN signatures extracted across primary tumors in a recent study [21]. Briefly, the tool explains the CN profile of each de novo signature as a linear combination of the profiles of one or several signatures in the reference set. In this case, the sigProfiler decomposes the 7 CN signatures extracted into linear combinations of 10 CN reference signatures, although the cosine similarity of the reconstructed and original de novo signatures is low in general (S3B Fig). In any case, no significant association (using the approach described above) exists between the activity of any of these CN reference signatures and with the exposure to any anticancer therapy (S3C Fig).

## Platinum CN footprint and distribution along the genome

To integrate all CN categories with different prevalence between platinum exposed and unexposed tumors into a platinum CN footprint, we first combined the prevalence (count) of chromosomal fragments in each CN category across all platinum exposed and unexposed WGD tumors, irrespective of their tumor type. Then, we summed over each CN feature the prevalence observed for exposed tumors, and separately that observed across unexposed tumors. Finally, we subtracted the vector representing the cumulative platinum CN profile of unexposed tumors from the vector representing the cumulative platinum CN profile of exposed

tumors. Thus, in the resulting vector, positive components correspond to CN categories with higher prevalence across platinum exposed tumors. We selected these to form the platinum CN footprint (Fig 3A). For practical purposes, to directly measure the intensity of this footprint in tumors, we directly count the number of heterozygous or LoH chromosomal fragments with CN between 1 and 4, and of length smaller than 10 Mb. This definition was used to study the distribution of the intensity of the platinum CN footprint along the genome (see below), to compare that intensity in treated/untreated and WGD/non-WGD tumors (Fig 4A) and to study its correlation with the platinum SBS footprint (Fig 4B).

To determine the intensity of the platinum CN footprint along the genome, we divided the genome into regions of 1Mb and calculated the intensity of the CN footprint per WGD tumor in each region Then, we subtracted intensity in each region for platinum non-exposed tumors from that of platinum exposed tumors. The resulting exposed-to-unexposed differences in intensity per genomic region are presented in Fig 3B.

### Correlation between platinum SBS signature and CN footprint

We extracted a new set of SBS signatures on its 3 base nucleotide context (SBS96) across HMF and PCAWG tumors. In order to increase the number of samples from the same tumor type and to perform a more accurate attribution of signatures, we aggregated metastatic and primary samples from the same cancer types. The matrix of counts of SBS in their trinucleotide contexts (SBS96 context) was constructed using SigprofilerMatrixGenerator version 1.1. [32], and the extraction of signature profiles and their activities across tumors was carried out using the SigProfilerExtractor version 1.1 [33] for a minimum of 2 and a maximum of 20 signatures per tumor type.

The final signatures' extraction result corresponds to the one recommended by the SigProfilerExtractor. Across the suggested sets of signatures, the platinum signature was selected as the one with a cosine similarity higher than 0.9 with any of the previously described platinum signatures (SBS31 and SBS35) in COSMIC version 3.2.

The presence of the platinum SBS signature on exposed samples is limited to those tumors that underwent a clonal expansion fixing the mutations of the treatment [6,8,11]. We selected exposed samples with platinum SBS signature activity.

Finally, we compared the number of mutations represented by that activity with the intensity of the platinum CN footprint (see above). The samples were divided into tertiles according to their amount of detected platinum SBS (quantiles 0.33, 0.66, 1) and the significant differences in the intensity of the platinum CN footprint across groups was estimated with a two-tailed Wilcoxon-Mann-Whitney test.

### Platinum survival model

For each of the patients, the time to death represents the time between the date of the annotated biopsy and the date of death. Those patients without a date of death are considered alive and the last follow-up is fixed at the date when the current version of the data was published (October 21th, 2021). The survival analysis follows a Cox proportional hazards regression model for each tumor type of 4 different groups of samples: (I) Platinum-exposed with WGD, (II) Platinum-exposed without WGD, (III) Platinum-unexposed with WGD, and (IV) Platinum-unexposed without WGD.

### Analysis of POG570 cohort

We obtained files containing the clinical and treatment data and CN calls of 570 metastatic tumors in the POG570 cohort [10] from a repository set up by the team that sequenced them (https://www.bcgsc.ca/downloads/POG570/). Patients who received platinum-based drugs

(cisplatin, oxaliplatin and/or carboplatin) as part of the treatment of the primary tumor were identified. Only the colon, breast or lung cohorts satisfied the requirement of at least 10 platinum exposed and unexposed tumors, and were thus selected for analysis. The length of chromosomal fragments identified in each tumor were computed as the difference between their start and end genomic coordinates. Then, the chromosomal fragments of CN 1–4 and length below 10 Mb in each tumor (platinum CN footprint) were counted and the counts of exposed and unexposed tumors compared. To compute the number of days elapsed between the end of the platinum treatment and the biopsy across exposed tumors, the relative date of treatment end was subtracted from the relative biopsy date.

## Supporting information

**S1 Fig.** Primary and metastatic WGD tumors a) Separation between WGD and non WGD tumors across the primary (PCAWG) and metastatic (HMF) cohorts on the basis of their ploidy and level of homozygosity. Similar to Fig 1A, but with the tumors of the two cohorts in separate panels. b-i) Distribution of ploidy (left) and fraction of the genome with LOH (right) of WGD tumors of different cancer types across primary and metastatic cohorts. Similar to Fig 1C, but separated by cancer types represented in both, PCAWG and HMF cohorts. The boxes inside the violin plots delimit the first, second and third quartiles of the distribution. All tumors in each group are represented as dots. P-values were derived from a two-tailed Wilcoxon-Mann-Whitney test.
(PDF)

**S2 Fig. Results of the comparison of CN categories across WGD ovarian, esophageal, breast triple negative and breast ER+/HER2- tumors exposed or unexposed to taxanes in the HMF cohort.** Significant comparisons (uncorrected p-value below 0.05) are highlighted with asterisks below each bar.
(PDF)

**S3 Fig.** Comparison with CN signatures a) Comparison of the activity of CN signatures extracted de novo from HMF WGD tumors between samples exposed and unexposed to different anticancer treatments. No signature appears with significantly different activity between exposed and unexposed WGD tumors. b) Equivalence (linear combination reconstruction) between CN signatures extracted de novo from the HMF cohort and CN signatures previously extracted from primary tumors (ref. 21). c) Comparison of the activity of CN signatures extracted from primary tumors (ref. 21) between samples exposed and unexposed to different anticancer treatments. No signature appears with significantly different activity between exposed and unexposed WGD tumors in the HMF cohort.
(PDF)

**S4 Fig. Platinum CN footprint distribution across chromosomes.** Number of chromosomal fragments with copy number 1–4 and length smaller than 10 Mb across WGD lung (a) and colorectal (b) tumors exposed or unexposed to platinum-based drugs. Individual points represent the number of chromosomal fragments in a chromosome in an exposed or unexposed tumor. P-values represent the significance of a two-tailed Wilcoxon-Mann-Whitney test.
(PDF)

**S5 Fig. Distribution of average ploidy of WGD tumors from different organs of origin that were exposed or unexposed to platinum-based therapies.** The distributions of both groups of tumors from each organ are compared using a two-tailed Wilcoxon-Mann-Whitney test.
(PDF)

**S6 Fig. Results of the comparison of CN categories across WGD urothelial and ovarian tumors exposed or unexposed to platinum-based therapies in the HMF cohort.**
(PDF)

**S7 Fig. Platinum CN footprint across tumors of an independent cohort (POG570).** a) Days elapsed between the end of platinum treatment and the biopsy of metastatic or recurrent tumors across the POG570 and HMF cohorts. Significantly longer time lapses for breast and lung tumors (and close to significant for colorectal tumors) are apparent across the HMF cohort. As a result, there is a higher likelihood of clonal expansion between treatment and biopsy (thus increasing the probability to detect platinum-related SBS and CN) in HMF metastatic tumors. This, together with the larger sample size probably explains why the platinum CN footprint is not as clear across POG570 colon tumors. One platinum exposed colorectal patient in the POG570 cohort did not have available CN data. b) Number of chromosomal fragments of size below 10 Mb identified across platinum-exposed and unexposed colon, breast and lung tumors. These numbers are significantly greater across exposed breast and lung tumors (two-tailed Wilcoxon-Mann-Whitney) and slightly greater (although not significant) across colon tumors. The platinum CN footprint is thus replicated in the POG570 cohort, except for colon tumors.
(PDF)

**S8 Fig. Mortality curves of patients bearing metastatic tumors originated in the lung and colorectal.** Each curve represents a group of patients bearing either platinum-exposed or unexposed and WGD or non WGD tumors. The p-values were obtained using a Kaplan-Meier test.
(PDF)

**S1 Table. Fraction of WGD tumors across PCAWG and HMF cancer types.**
(XLSX)

**S2 Table. Number of WGD and non-WGD metastatic tumors of each tumor type across the HMF cohort exposed to different anti-cancer therapies.**
(XLSX)

**S3 Table. Association of CN features of WGD tumors with anticancer treatments across the HMF cohort.**
(XLSX)

**S4 Table. Overlap of tumors exposed to pairs of anticancer therapies in different cancer types across the HMF cohort.**
(XLSX)

## Acknowledgments

The authors wish to acknowledge the contribution of patients, families and biomedical researchers who shared, processed and sequenced the data used within the study. The results published here are in part based on the data generated by the Pan-Cancer Analysis of Whole Genomes. This publication and the underlying study have been made possible partly on the basis of the data that Hartwig Medical Foundation has made available to the study. The authors acknowledge the contribution of Claudia Arnedo-Pac of signatures extraction in the HMF cohort. IRB Barcelona is a recipient of a Severo Ochoa Centre of Excellence Award from the Spanish Ministry of Economy and Competitiveness (MINECO; Government of Spain) and is supported by CERCA (Generalitat de Catalunya).

## Author Contributions

**Conceptualization:** Santiago Gonzalez, Nuria Lopez-Bigas, Abel Gonzalez-Perez.

**Data curation:** Santiago Gonzalez.

**Formal analysis:** Santiago Gonzalez.

**Investigation:** Santiago Gonzalez.

**Methodology:** Santiago Gonzalez, Abel Gonzalez-Perez.

**Software:** Santiago Gonzalez.

**Supervision:** Nuria Lopez-Bigas, Abel Gonzalez-Perez.

**Validation:** Santiago Gonzalez.

**Writing – original draft:** Santiago Gonzalez, Nuria Lopez-Bigas, Abel Gonzalez-Perez.

**Writing – review & editing:** Santiago Gonzalez, Nuria Lopez-Bigas, Abel Gonzalez-Perez.

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
