## [Decision Letter · Decision Letter 0]

24 Jan 2023

Dear Dr Gonzalez-Perez,

We are pleased to inform you that your manuscript entitled "Copy number footprints of platinum-based anticancer therapies" has been editorially accepted for publication in PLOS Genetics. Congratulations!

Please, note that Reviewer 3 had comments and detected apparent typos.  Please, address these in creating final version for publishing.

Yours sincerely,

Dmitry A. Gordenin, Ph.D.

Academic Editor

PLOS Genetics

David Kwiatkowski

Section Editor

PLOS Genetics

Comments from the reviewers (if applicable):

Reviewer's Responses to Questions

**Comments to the Authors:**

Reviewer #1: My previous comments have been adequately addressed with this manuscript revision.

Reviewer #2: Authors have responded to my comments satisfactorily.

Reviewer #3: This resubmission of the research by Gonzalez et al represents a thorough improvement on the previous work. Of particular note, the comparison to a new independent cohort, which although limited in size, reproduced the main finding. The comparison to Steele et al is also appreciated along with the addition of supplementary tables with sample information and the inclusion of sample numbers throughout the figures which improved the clarity of the paper.

I recommend publication of this improved manuscript.

Below are some minor typos which should be fixed prior to publication.

Some minor comments

Fig 1b cancer samples. Would be a nice comparison to know the differences in the cohorts here.

Fig 1c legend typo LoOH

Fig3a in figure variaton

Fig 4 legend

Twice says “with copy number 1-4 and length smaller than 10 Mbwith length between 10 Kb and 10 Mb”

**Have all data underlying the figures and results presented in the manuscript been provided?**

Reviewer #1: Yes

Reviewer #2: Yes

Reviewer #3: None

PLOS authors have the option to publish the peer review history of their article (what does this mean?). If published, this will include your full peer review and any attached files.

Reviewer #1: No

Reviewer #2: No

Reviewer #3: No

**Data Deposition**

http://datadryad.org/submit?journalID=pgenetics&manu=PGENETICS-D-23-00053

**Press Queries**

---

## [Editor Report · Acceptance letter]

3 Feb 2023

PGENETICS-D-23-00053 

Copy number footprints of platinum-based anticancer therapies 

Dear Dr Gonzalez-Perez, 

We are pleased to inform you that your manuscript entitled "Copy number footprints of platinum-based anticancer therapies" has been formally accepted for publication in PLOS Genetics! Your manuscript is now with our production department and you will be notified of the publication date in due course.

With kind regards,

Zsofia Freund

PLOS Genetics

On behalf of:
